

# Long-term development of a perennial firn aquifer on the Lomonosovfonna ice cap, Svalbard

Tim van den Akker[1], Ward van Pelt[2], Rickard Petterson[2], Veijo A. Pohjola[2]

[1]Institute for Marine and Atmospheric Research Utrecht, Utrecht University, Netherlands

[2]Department of Earth Sciences, Uppsala University, Uppsala, Sweden

*Correspondence to*: Tim van den Akker (t.vandenakker@uu.nl)

**Abstract.** An uncertain factor in assessing future sea level rise is the melt water runoff buffering capacity of snow and firn on glaciers and ice caps. Field studies have resulted in observations of perennial firn aquifers (PFAs), which are bodies of water present deep in the firn layer and sheltered from cold surface conditions. PFAs can store surface melt, thereby acting as a buffer for sea level rise, and influence the thermodynamics of the firn layer. Furthermore, ice dynamics might be affected by the presence of liquid water through hydrofracturing and water transport to the bed. In this study, we present results of applying

an existing groundwater model MODLFOW 6 to an observed perennial firn aquifer on the Lomonosovfonna ice cap in central Svalbard. The observations span a three-year period, where a ground penetrating radar was used to measure the water table depth of the aquifer. We calibrate our model against observations to infer hydraulic conductivity $6.4 * 10^{-4}$ m s$^{-1}$, and then use the model to project the aquifer evolution over the period 1957 – 2019. We find that the aquifer was present in 1957, and that it steadily grew over the modelled period with relative increases of about 11 % in total water content and 15 % in water table

depth. Water table depth is found to be more sensitive to transient meltwater input than firn density changes at this location on the long term. On an annual basis, the aquifer exhibits sharp water table increases during the melt season, followed by slow seepage through the cold season.

## 1 Introduction

Aquifers, defined in groundwater analysis as an underground layer of permeable material that trap liquids in its pore space,

can hold substantial volumes of water. Firn, defined as multi-year snow with a lower density than ice, contains pore spaces and can therefore also contain water. Kuipers Munneke et al. (2014) found that aquifers can form in firn layers of ice caps or glaciers, given certain meteorological conditions. If a firn aquifer persists for multiple years, it is referred to as a perennial firn aquifer (PFA).

PFA's can form when i) there is enough water input into the firn, either by rain or surface melting and ii) there is enough pore

space at depth to accommodate the percolated water and to shelter the PFA from refreezing during (winter) cold periods



(Kuipers Munneke et al., 2014). Future climate warming introduces a positive and negative effect on PFA extent. On the one hand, increased (atmospheric) temperatures lead to more meltwater and/or rain as input to a PFA, and a less cold winter period reduces conductive cooling and refreezing of a PFA from above. On the other hand, higher firn temperatures lead to a denser firn package (Kuipers Munneke et al., 2014, Van Pelt et al., 2019, Brils et al., 2022), and less firn air content or pore space

(Veldhuijsen et al., 2023), lowering the potential of the firn to retain meltwater and thereby preventing the aquifer from growing. Regions in which PFA's are already present might disappear or move up-glacier once the firn is too dense to foster a PFA, such as on the Greenland Ice Sheet and Svalbard. Other areas, that where too cold for PFA formation in the past but have a deep and airy firn layer might turn to PFA locations. This could happen on the Antarctic Ice Sheet (Veldhuijsen et al., 2024).

A key requirement for PFAs to form is temperate conditions at depths $>\sim10$ m below the surface (in addition to a lack of options for the water to drain via crevasses). In Svalbard, temperate firn at such depths exists across nearly all accumulation zones, and previous work has already shown the existence on the Holtedahlfonna ice cap (Christianson et al., 2015) and the the Lomonosovfonna ice cap (Hawrylak and Nilsson, 2019). Before this, PFAs have been found in Greenland (Forster et al., 2014, Koenig et al., 2014) and on mountain glaciers in other Arctic regions, e.g. in Canada (Ochwat et al., 2021) or the USA

(Fountain, 1989). Miller et al. (2020) found that the residence time of water in the Greenland PFA is about 6,5 years, and Poinar et al. (2017) found that aquifer flow enhanced the formation of deeper crevasses. Recently, surface melt water streams have been identified on Antarctica that are said to be able to form firn aquifers (Kingslake et al., 2017). Modelling studies have already shown that perennial firn aquifers can form on the Antarctic peninsula (van Wessem et al., 2021) and on the rest of the Antarctic Ice Sheet (van Wessem et al., 2020). Radar observations of an East Antarctic outlet glacier indicate the possible

presence of a PFA, comparable to the one found on the GrIS (Lenaerts et al., 2017, Lenaerts et al., 2018, Schaap et al., 2020).

Efforts have been made to model the formation and development of PFAs. Kuipers Munneke et al. (2015)present a one-dimensional aquifer model, which is designed to assess under what climatic conditions an aquifer will form, and how wet firn responds to climate change, compared to dry firn. Efforts have also been made to assess the capability of different models to predict vertical meltwater percolation (Marchenko et al., 2017, Steger et al., 2017). Vertical percolation and refreezing is more

elaborately solved in the 1D Crocus/SURFEX model (Vionnet et al., 2012), and also the Richardson Equation is used to model vertical water flow (Wever et al., 2014). More recently, Miller et al. (2023) modelled a perennial firn aquifer on the Helheim glacier in Greenland using SUTRA-ICE, a 2D model combining groundwater flow with a subsurface energy balance model to calculate freeze-thaw cycles. For an extensive comparison of firn models and their components, the reader is referred to Vandecrux et al. (2020) and Stevens et al. (2020).

A key parameter needed to accurately model a PFA is the hydraulic conductivity of firn and snow, being the main control on water flow rates. Fountain and Walder (1998) examined field tests of five different glaciers and found a range of 1 - 5 $*10^{-5}$ m



s$^{-1}$. They argue that firn conditions are therefore uniform between glaciers, because of the low range in measured hydraulic conductivity. Miller et al. (2017) did slug tests on the Greenland Ice Sheet (GrIS) and found hydraulic conductivities ranging between $2.5 *10^{-5}$ m s$^{-1}$ and $1.1 *10^{-3}$ m s$^{-1}$. Stevens et al. (2018) found in their literature review a range of $10^{-6}$ - $10^{-2}$ m s$^{-1}$,
using similar techniques as Miller et al. (2017) on ten northern hemisphere glaciers (in Canada, Svalbard, northern Sweden, Greenland and the Alps).

In this study, we combine three years of field data, a state-of-the-art firn model and a 3D groundwater-flow model to simulate the evolution of a PFA found on the Lomonosovfonna ice cap on Svalbard from 1957 - 2019. We approach the PFA as a classical groundwater system of flow through a porous medium, with meltwater input at the top and outflow through the
boundaries. Surface and firn processes such as melt, water percolation, refreezing, heat diffusion and firn densification are resolved by the Energy Balance Firn Model (EBFM; Van Pelt et al. (2019)), which provides input to the PFA model. The model is calibrated using the hydraulic conductivity and parameters related to the storage of water to fit with in-situ observations of three consecutive years, obtained with ground-penetrating radar (GPR) measurements. Our objective with this work is to add to the knowledge of the dynamics of firn aquifers, and in particular how changes in the aquifer can be described
numerically by a hydrogeological model. Through model calibration we infer hydraulic conductivity. Furthermore, the modelling gives a long-term perspective of how the thermal regime and water storage have changed on an Arctic ice cap since the 1950s.

## 2 Study area

The Lomonosovfonna ice cap is situated in central eastern Spitsbergen, the largest island of the Svalbard-archipelago, see
Figure 1. It is the highest ice cap of Svalbard, reaching up to 1250 m a.s.l. The Lomonosovfonna ice cap is about 600 km$^2$, and feeds into several outlet glaciers. Two of these outlet glaciers, Tunabreen and Negribreen have surged over the last 20 years (Isaksson et al., 2001, Flink et al., 2015, Haga et al., 2020). Nordenskiöldbreen and the Lomonosovfonna ice cap are monitored since 1997 by Uppsala University and Utrecht University (Van de Wal et al., 2002, Marchenko et al., 2017, Marchenko et al., 2019, Marchenko et al., 2021). The monitoring program consists of mass balance monitoring, ice velocity measurements, ice
thickness measurements and meteorological observations. A map of the Lomonosovfonna, with elevation lines and the locations where the observations were done, is presented in Figure 1b. Previous glaciological studies, using the data from the monitoring program, on Nordenskiöldbreen and the Lomonosovfonna ice cap have assessed climatic mass balance (Van Pelt et al., 2012), snow and firn conditions (Pohjola et al., 2002, Marchenko et al., 2017, Marchenko et al., 2019) and ice dynamics and thickness (Van Pelt et al., 2013, Van Pelt et al., 2018). The modelled grid in this study, together with the areas where GPR
observations were made per year, is shown in Figure 1b.



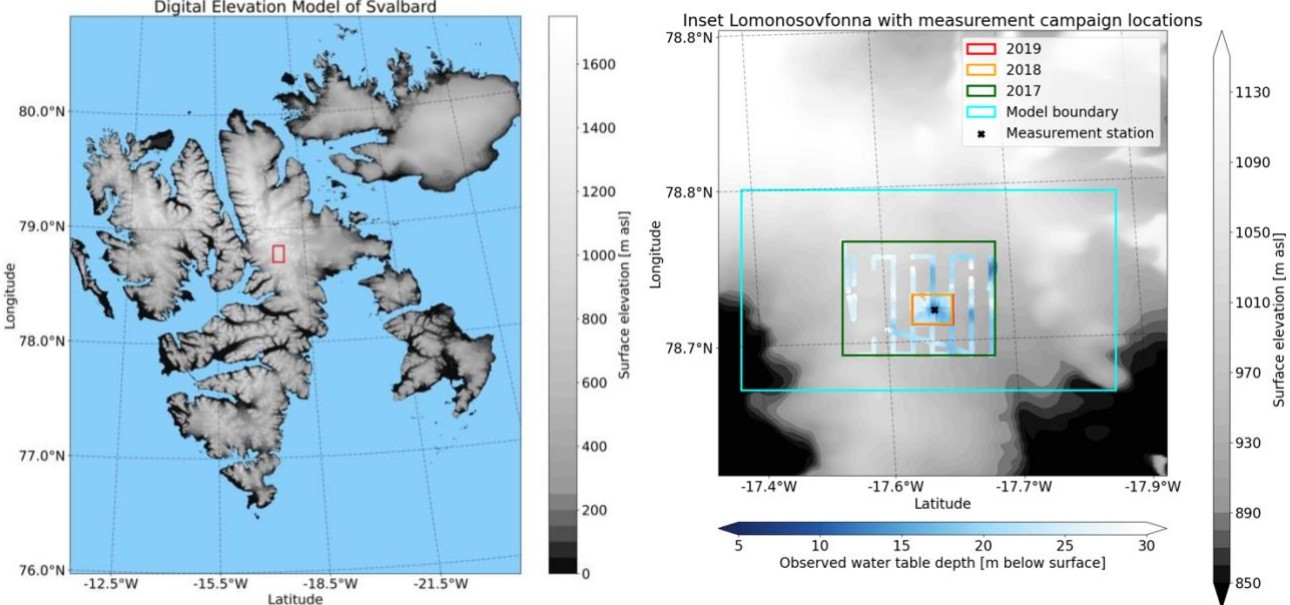

**Figure 1 (left).** The Svalbard archipelago, plotted from the digital elevation model of Melvær et al. (2014). The red rectangle shows the inset shown on the (right), which is the location of the measurements over the years 2015 (purple), 2016 (blue), 2017 (yellow), 2018 (orange) and 2019 (red). The rectangles correspond to the minimum and maximum coordinates of the measurement in those years and thus show the extend of the PFA measurements. The datapoints obtained in these campaigns are turned to water depths (using the methods described in the next paragraph). The datapoints and extend of the measurement campaign of the years 2018 and 2019 overlap. The measurement station operated by the Uppsala University is shown by a black cross in the middle of the 2018 and 2019 measurement campaigns.

## 3 Data and methods

### 3.1 Water table height data

We use water table depth measurements obtained at the Lomonosovfonna ice cap as our main tuning data for the ground water flow model. The water table depth was obtained from ground penetrating radar data, where the water table stands out clearly due to the large permittivity difference between liquid water and air-saturated firn (Fig. 2). The GPR data was collected with a Malå ProEx radar system with a 250 MHz antenna pulled behind a snowmobile. The positioning of the radar profiles was done with a two-frequency geodetic GNSS receiver. The GPR profiles followed a grid pattern over the study area (Figure 1) and were collected during the spring of 2017, 2018 and 2019. The typical overall length of the radar profiles during one season is roughly 40 km.



The raw GPR data was minimally processed with zero-time adjustment and a low-pass filter (300 MHz cut-off frequency). An
example radargram is shown in Figure 2. The two-way travel times (TWTT) of the radar signal to the reflective surface of the
water table were manually digitized and converted to depth using a vertical profile of interval signal velocities in the firn. The
dielectric constant of the layers were determined from a firn density profile following Kovacs et al. (1995) where $\rho_f$ is the
density of the firn layer and $\rho_w$ the density of water:

$$\epsilon_f = \left( 1 + 0.845 * \frac{\rho_f}{\rho_w} \right)^2 \tag{1}$$

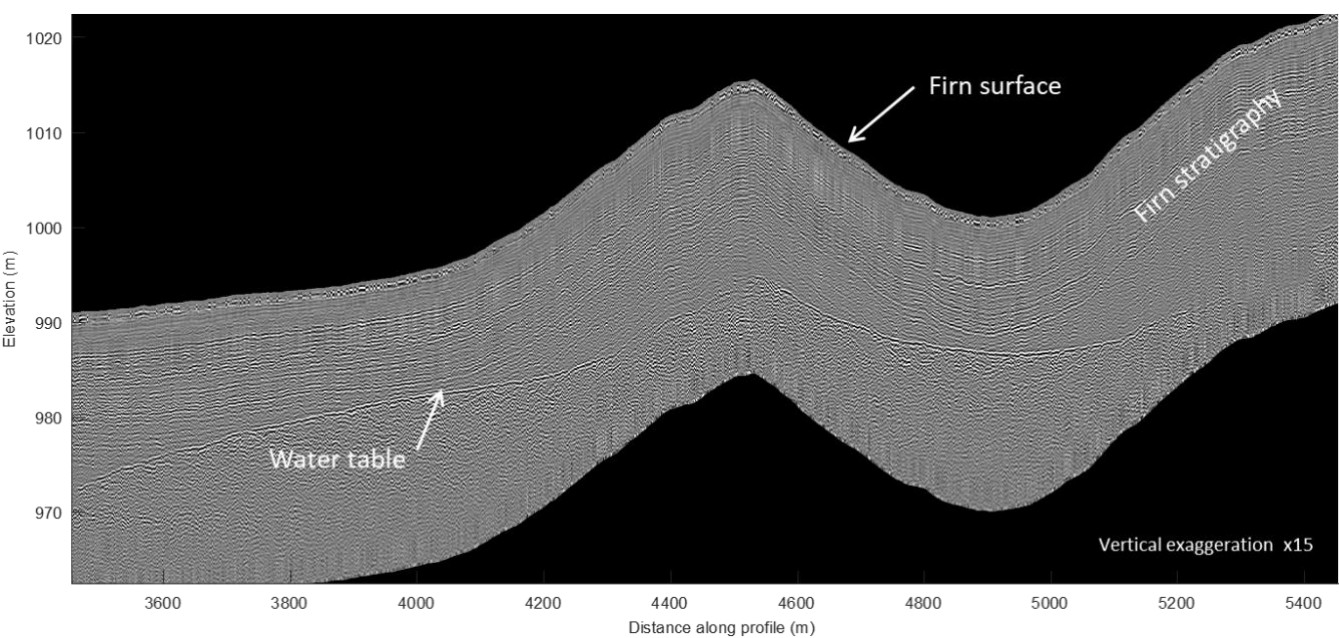

**Figure 2:** An example of topographical corrected (x15 exaggeration) radargram. The water table of the firn aquifer is clearly visible and
cuts the general firn stratigraphy. The resulting observed water table depth, plotted over the contour lines of the surface elevation of the
model grid, can be observed in Figure 1.

### 3.2 Water input & firn density

Firn density and meltwater input to the aquifer is required to model a PFA. This is retrieved from the Energy Balance - Firn
Model (EBFM) (Van Pelt et al., 2012, Van Pelt and Kohler, 2015, Marchenko et al., 2019). Here, we use output from the
model version of the EBFM presented in Van Pelt et al. (2019), which has previously been calibrated and validated against
stake measurements, weather station data and subsurface measurements across Svalbard. A summary of EBFM components
that are most relevant for this study is given below.






EBFM first solves the surface energy balance equation:

$$Q_{melt} = SW_{net} + LW_{net} + Q_{sens} + Q_{lat} + Q_{sub} \tag{2}$$

Energy fluxes at the surface that are part of this model are net shortwave radiation flux ($SW_{net}$), net longwave radiation flux ($LW_{net}$), turbulent heat exchange flux through latent and sensible heat (respectively $Q_{lat}$ and $Q_{sens}$), and the conductive heat flux into the ice ($Q_{sub}$). Equation (2) is solved for the surface temperature with a melt energy flux ($Q_{melt}$) set to zero. If the resulting surface temperature is higher than 0 °C, the surface temperature is set at 0 °C and the energy fluxes in equation (2) are recomputed, resulting in a value for $Q_{melt}$>0. EBFM requires air temperature, pressure, relative humidity, cloud cover and precipitation as input to solve the surface energy balance, which comes from downscaled HIRLAM NORA10 regional climate model output (Reistad et al., 2011).

Below the surface, the EBFM solves the thermodynamic equation and the densification equation. The thermodynamic equation, in which ρ is the firn density, $c_p(T)$ the temperature dependent heat capacity of the firn, κ(ρ) the density dependent effective conductivity of the firn, F the refreezing rate (in kg m$^{-3}$ s$^{-1}$), $L_m$ the latent heat of melting ice (3.34 x 10$^5$ J kg$^{-1}$) and z the vertical coordinate (Δz is the layer thickness), describes the change of the temperature-depth profile due to heat conduction and refreezing of the meltwater resulting from the surface energy balance.

$$\rho c_p(T)\frac{\delta T}{\delta t} = \frac{\delta}{\delta z}\left(\kappa(\rho)\frac{\delta T}{\delta z}\right) + FL_m \tag{3}$$

The densification equation describes the change of density of the firn due to gravitational compaction and refreezing.

$$\frac{\delta \rho}{\delta t} = K_g(\rho, T) + \frac{F}{\Delta z} \tag{4}$$

Here, $K_g(\rho,T)$ describes density and temperature dependent compaction. The density of fresh deposited snow is fixed at 350 kg m$^{-3}$. Gravitational compaction $K_g(\rho,T)$ follows Ligtenberg et al. (2011):

$$K_g(\rho, T) = C(b)bg(\rho_{ice} - \rho)\,exp\left(-\frac{E_c}{RT} + \frac{E_g}{RT_{avg}}\right) \tag{5}$$

in which C(b) is an accumulation dependent parameter (Ligtenberg et al., 2011), b the accumulation rate, $\rho_{ice}$ the density of ice (typically 917 kg m$^{-3}$), Ec the activation energy of creep by lattice diffusion (typically 60 kJ mol$^{-1}$), Eg the activation energy of gain growth (typically 42.4 kJ mol$^{-1}$), R the gas constant, T the temperature of the firn and $T_{avg}$ the temporal mean subsurface temperature..

Water in the EBFM originates from surface melt and rain, and percolates down from the surface into the firn. First, the water refreezes when the conditions in a model layer are sufficient, being that the temperature should be below the melting point and



the density should be lower than the density of ice. Refreezing raises the temperature and density. If not all water refreezes, a
small portion will be stored in the layer as irreducible water content. The remaining water will percolate down to the next
layer, where the process repeats. This continues until the water encounters a layer that has the density of ice, where it will pile
up to fill the pore spaces of the firn above the ice. Furthermore, fast deep percolation is modelled using the parameterization
by Marchenko et al. (2017). All percolating melt and rain water that is not stored or refrozen in snow and firn is assumed to
be the water input source to the PFA.

## 3.2 Firn aquifer modelling

The US Geological Survey (USGS) Modular Hydrological Model MODFLOW 6 is chosen in this research to model the
horizontal and vertical water flows in the PFA on the Lomonosovfonna ice cap. For an extensive documentation, the reader is
referred to (Bakker et al., 2016, Langevin et al., 2017)).

Liquid flow in porous media is governed by Darcy's law, which states that a liquid will flow from areas with a higher water
surface elevation (often called head height, water table or water table height, here we will use the latter) towards areas with a
lower surface water elevation. It is found experimentally by Miller et al. (2020), by using salt injection in boreholes in an
aquifer on the GrIS, that water flow in a firn aquifer generally obeys Darcy's Law and can therefore be approached as a
groundwater flow system. The simplest form of the Darcy equation can be observed in equation 6, in which q is the flow per
unit area [in m s$^{-1}$], k the hydraulic permeability [in m$^2$], μ the dynamic viscosity of the fluid [in Pa s] and ∇p the pressure
gradient vector [Pa m$^{-1}$].

$$q = -\frac{k}{\mu}\nabla p \qquad (6)$$

The hydraulic permeability is a measure of how easy a fluid moves through a medium. A higher k indicates less resistance
from the medium to the flow. Often, hydraulic conductivity and hydraulic permeability are interchangeably used. Hydraulic
conductivity and hydraulic permeability are linked in equation 7, in which K is the hydraulic conductivity, k the hydraulic
permeability, ρ the density of the fluid, g the gravitational acceleration and μ the viscosity of the fluid.

$$K = \frac{k\rho g}{\mu} \qquad (7)$$

Equation 6 assumes that the hydraulic permeability remains spatially constant, and all sources and sinks of the water are
summarized in the ∇p term. A more comprehensive Darcy equation, adapted from Langevin et al. (2017), is:



$$\frac{\delta}{\delta x}\left(K_{xx}\frac{\delta h}{\delta x}\right) + \frac{\delta}{\delta y}\left(K_{yy}\frac{\delta h}{\delta y}\right) + \frac{\delta}{\delta z}\left(K_{zz}\frac{\delta h}{\delta z}\right) = SS\frac{\delta h}{\delta t} + Q_S \tag{8}$$

The first three terms on the left-hand side of equation 5 represent the flow due to differences in water height, with a spatially varying and direction-dependent hydraulic conductivity. $Q_S$ represents different sources and sinks, such as surface runoff,

evapotranspiration, wells and precipitation. Note that those sinks and sources can be space and time dependent. The SS term on the right-hand side refers to specific storage, which is the water released or stored per drop of head from the pore storage. $K_{xx}$, $K_{yy}$ and $K_{zz}$ are the hydraulic conductivities that control the speed of the water flow in respectively the x, y and z direction.

The standard packages that regulate the density are time independent in MODFLOW 6, which means that density typically does not change during a (regular) MODFLOW simulation. In groundwater modelling, with for example solid porous media like rocks and sand, this can be a justifiable simplification. This is, in the case of modelling a PFA, not desired, as the density

changes significantly during a simulation.

It is therefore needed to make separate MODFLOW iterations every time step the steady packages need to change. To connect these separate MODFLOW iterations into one model, the head heights of run t are used as initial conditions for run t+1. This allows steady packages to change per time step, but also to make more efficient use of storage. This makes the model slower

compared to a default MODFLOW 6 run. However, all runs in this study take approximately 3 hours to run on a single processor (approximately 60 years with weekly resolution). A schematic overview of the modelling process is presented in Figure 3.



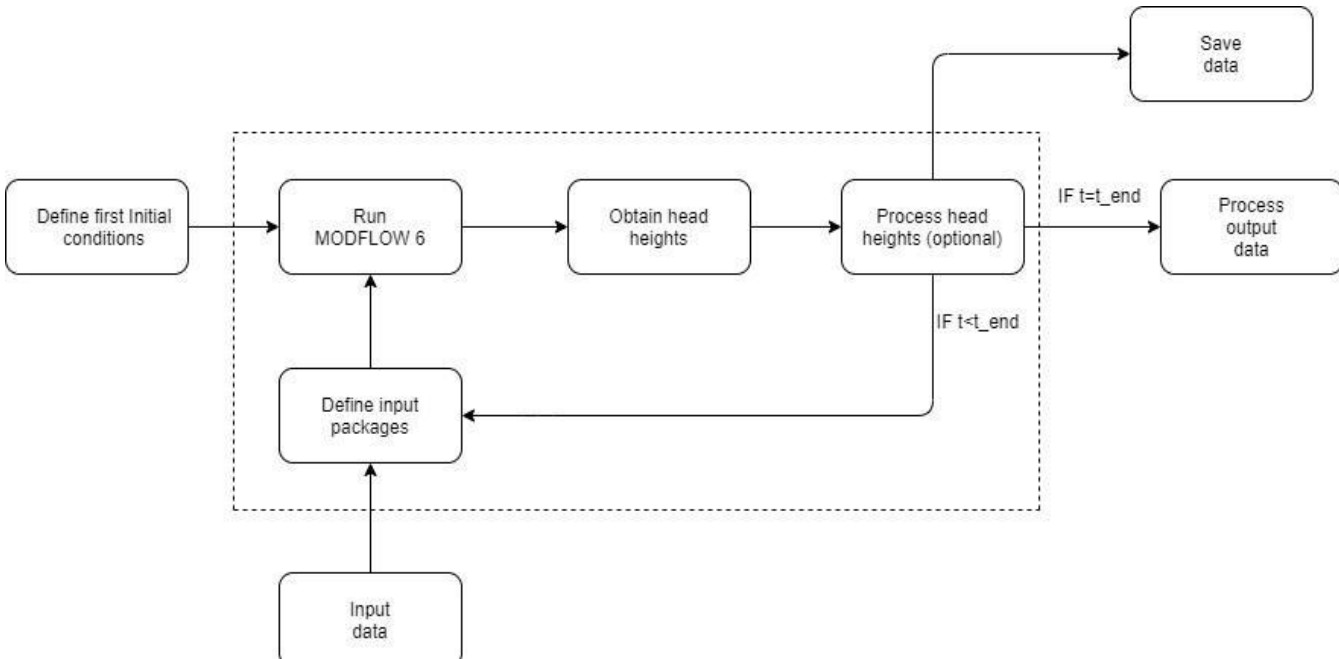

**Figure 3:** The modelling process, in which every time step a new MODFLOW model is used, using the water table height from the previous
time step as initial conditions. The dotted lines indicate the loop, for which every loop is a new MODFLOW model, that uses the head height
output of the previous loop as initial conditions. Note that input data can be read in every loop, instead of all at once, and that output data
can be stored after each loop.

We run the Lomonosovfonna Perennial Firn Aquifer Model (LPFAM) on a grid of 100 by 100 cells, with 96 meters resolution
in the y direction and 72 meters in the x direction. The uneven grid discretization originates from the selection of the area of
interest and a buffer zone around it, to minimize the effect of boundary conditions, as well as the vertical resolution of the
input datasets used. The vertical discretization is done in 5 layers with increasing depth, for the density taken from the EBFM.
However, the water table height is solved on a sub-grid scale, so on a much higher vertical resolution. We choose the domain
such that there is a buffer zone of several kilometers between the largest observational grid with reasonable sampling (2017,
see Figure 1) and that the grids with the most datapoints (2018 and 2019) are roughly in the middle of the grid, to minimize
effects of the boundary conditions. At the start of a spin-up routine, we fix the water table height at 10 meters above the bottom
of the deepest model layer at the boundaries. This is the depth at which the firn density reaches the density of glacial ice. This
varies spatially, and is typically around 50 meters below the surface. We found no influence on the modelled water table height
in 2017 of either choosing 5,10,15 and 20 meters as initial water table height. The modelled PFA responds differently to
different initial conditions in the first decade after 1957.



We performed a spin-up of 20 years, where we run 1957-1977, with in 1957 an arbitrary initialized state of 10 meters water column above the bottom of the model domain. The resulting water table profile is used as initial condition for an historical run from 1957 to 2019. We manually tune uncertain parameters to match the GPR observations in the years 2017 – 2019, described in the next section. We redo every spin-up (1957 – 1977) with the new parameter values chosen for completeness, 210 and to avoid any initial shocks.

There are two available sources of surface elevation data: 1) the DEM of the Norwegian Polar Institute (NPI) and 2) surface altitude data from a GPS logger for the years 2017 - 2019, pulled together with the GPR over the grid shown in Figure 1. To be consistent, the elevation data from 2) is used to transfer the depth of the water table to head height $h$ in m a.s.l. , which is 215 the common output metric of MODFLOW 6. However, the modelled grid is larger than the observed grid. Therefore, for model calculations and discretization, the NPI DEM will be used, as this dataset has a high resolution (approximately 1-10 meters) and is available for the whole grid. The DEM surface elevation data on the points where the observations were made did not differ significantly with surface elevation data obtained by the GPS.

## 4 Results & Discussion

### 4.1 Model calibration & inference of hydraulic properties

In this study, we performed three steps to tune the LPFAM to reproduce a water table close to observations of the years 2017 - 2019. We manually tuned three uncertain parameters, starting with the parameter that caused the most significant changes in the water table. We quantified the quality of the tuned water table by calculating the RMSE and the bias with our observations.

225 First, we took a uniform hydraulic conductivity throughout the grid and we varied it between $10^{-5} - 10^{-3}$ m/s in 100 steps, representing the range of values found in the literature. The found uniform hydraulic conductivity is referred to as $K_u$. Then, we introduced a pore close-off depth, below which we expect the water not to be able to move horizontally. The firn density increase with depth, and as a consequenc, the pore space decrease with depth. Therefore it is less likely that water present in these pores can find its way to the next (Koenig et al., 2014, Humphrey et al., 2021). We varied the pore close-off depth on a 230 layer basis: we tried putting the close off depth at the 5th, 4th and 3rd layer. We modelled the pore close-depth with an extremely low hydraulic conductivity below it. However, it is still expected that the water can dissipate with an unknown rate from this layer, for example by entering crevasses, or slow horizontal flow. This process is not explicitly modelled, so we used the pore close-off hydraulic conductivity as a final tuning parameter. We varied the pore close-off hydraulic conductivity to be between $10^{-1} - 10^{-6}$ times the hydraulic conductivity found in the first tuning step. The found pore close-off hydraulic 235 conductivity is referred to as $K_c$. The optimal set of parameters is presented in table 1. A summary of performance statistics for the years with observations is shown in table 2. On average we assume a close off density of approximately 804 kg m$^{-3}$, happening at a depth of 52.9 m below the surface. Our pore close-off density is well within the range of field studies (Gregory et al., 2014) and other modelling studies (e.g. (Herron and Langway, 1980, Ligtenberg et al., 2011, Huss, 2013, Brils et al.,



2022), that typically fix the pore close-off density at 800-830 kg m$^{-3}$. Our hydraulic conductivity is also in the range of $1 * 10^{-3} - 1 * 10^{-5}$ [m s$^{-1}$] found by Miller et al. (2017), Miller et al. (2020), Miller et al. (2023) and (Stevens et al., 2018).

Table 1. Optimal values found in the three consecutive tuning steps

| Steps | Optimal value | Range tested and steps used |
|---|---|---|
| Step 1: Tuning the uniform hydraulic conductivity K | $6.4 * 10^{-4}$ [m s$^{-1}$] | $1 * 10^{-3} - 1 * 10^{-5}$ [m s$^{-1}$], in 100 steps |
| Step 2: pore close-off depth | Layer 5 | Layers 5,4,3 |
| Step 3: pore-close off seepage rate | K (step 1) $* 10^{-2}$ | $10^{1}-10^{-10}$, in 10 steps |

## 4.2 Validation of modelled water table depth

Figure 4 show the modelled water table and the error with observations side by side, computed for every observed location by matching to the closest modelled grid point. The LPFAM maximum vertical depth (~50 m, dependent on the thickness of the firn package) is deeper than what the GPR can observe (up to ~40 m). All radar observations with no detected water table are excluded from the comparison and show up as interruptions in the tracks in Figure 4, which does not mean that the water table is absent.

In 2017, observations covered the largest grid (Fig. 1b). The modelled water table is overestimated in the south-eastern corner of the grid. Although the spatial pattern is very comparable (R = 0.68), the LPFAM has a water table significantly closer to the surface (bias = 2.3 m). This could be because there are missing sinks in the LPFAM in that region. During observational campaigns, the GPR could not measure beyond the southwestern corner because of the presence of a crevasse field, see for example Hawrylak (2021). These crevasses could very well act as a sink that is currently unaccounted for in the model. As a result, the root-mean-square error (RMSE) is significantly larger for the large grid observed in 2017 (5.05 m) than for the small grid in 2018 (1.03 m) and 2019 (1.49 m), where vertical drainage is likely absent in the observational grid of those years. The underestimation of water table depth in the southeastern corner explains a mean bias of 2.53 m.

The modelled water table height agrees very well with our in situ observations on a smaller grid during the years 2018 (RMSE = 1.03 m, bias = 0.5 m, R = 0.97) and 2019 (RMSE = 1.49 m; bias = 1.23 m; R = 0.97). The north-western, south-western, and





north-eastern corners of the smaller modelled domain contain little water according to model and observations, whereas water
265 flow is steered by the surface topography to the flatter central part and south-eastern corner, where it leaves the observational
grid and moves into the buffer zone (see Figure 1b.).

In Figure 5, the modelled water table depth and the water table depth at the closest observation coordinate are shown. The
modelled water table pattern in 2018 and 2019 matches almost perfectly with the observations, highlighted by the correlation
270 coefficient close to 1. In 2017 the outliers in the modelled water table are mostly below the correlation line i.e. the modelled
water table is overestimated compared to the observations, confirming the missing sinks in the southwestern corner of the
model grid. Adding sinks would lower the modelled water table and remove the outliers.

From the height contours and water table height pattern it can be inferred that water table height variations are steered by
275 surface topography, rather than lateral firn density variations. Elevated (convex) areas are often modelled and observed as drier
locations, whereas (concave) dips in the terrain are associated with a water table close to the surface. This applies to both the
smaller grid (2018 and 2019) and the larger grid (2017).





**Figure 4.** (left column) modelled water table depth for the years with the most observational coverage (2017,2018,2019). The observational data point locations are shown in red. The surface height contours, taken from the DEM of the NPI per 10 meters are shown with black dashed lines. (right) The difference between the observed water table depth and the modelled water table depth, which is interpolated to the locations of the observations. The performance statistics are shown in table 2.





**Table 2.** Performance indicators of the perennial firn aquifer depth from the model and compared to observed firn aquifer depth. The RMSE
and bias are determined by comparing all modelled points within the observational grid of a given year with the closest observed location.
This results in 145, 143 and 835 datapoints respectively in the years 2019, 2018, and 2017. The mean modelled and mean observed depths
are the arithmetic means of the aforementioned matched data points

| Year | Mean modelled PFA depth [m] | Mean observed PFA depth [m] | RMSE [m] | Bias [m] | Correlation R [-] | P-value [-] |
|------|------|------|------|------|------|------|
| 2019 | 20.2 | 21.4 | 1.49 | 1.2 | 0.974 | < 0.001 |
| 2018 | 20.3 | 20.8 | 1.03 | 0.5 | 0,972 | < 0.001 |
| 2017 | 20.3 | 22.6 | 5.05 | 2.3 | 0.678 | < 0.001 |

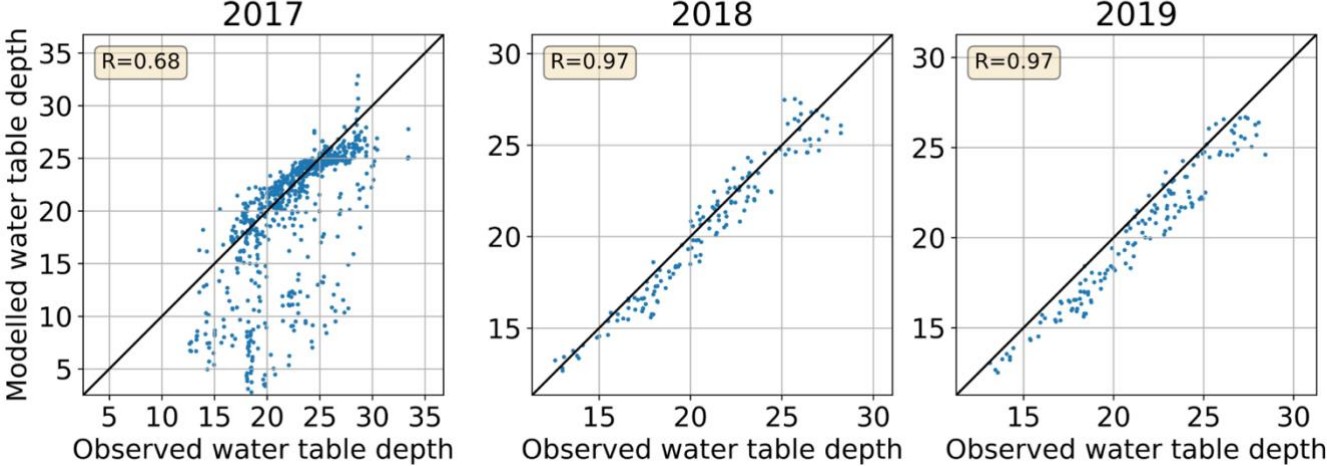

**Figure 5.** Scatter plots of the modelled water table and the closest observed location of 2017 (left), 2018 (middle) and 2019 (right). The
black line represents the R=1 full correlation relation, and the calculated correlation is shown in the top left corner.

## 4.3 Multidecadal PFA development

At the end of our spin-up routine in 1977 a PFA is present in the firn. Given that temperatures during 1957-1977 were likely
cooler than during 1937-1957 (Nordli et al., 2020), we argue it is highly likely that a PFA was already present in 1957 and in
the preceding decades. It is worth noting though that when starting without a PFA in 1957 it would also be possible to reproduce
present-day water table depths due to the short response time of the PFA relative to the simulation period. However, when
starting a simulation with no perennial firn aquifer present, we find an unphysical increase in water table depth even during



the cold season for the first ten years of the simulation (not shown). This modelled behaviour and the temperature dataset of Nordli et al. (2020) makes it very likely that the PFA was already present well before 1957.

In Figure 6, the modelled average water table over the whole grid is shown (including the buffer zone, see Fig 1). There is in general a positive trend visible, both in modelled average water table depth and meltwater input. The arctic has warmed during
this period (Van Pelt et al., 2019), and increased temperatures lead to more meltwater input and densification. Both processes decrease the modelled water table depth. There are periods visible in the average water depth where the water table seems to remain stable with interannual variability (1957-1970, 1970-2000, 2000-2010 and 2010-2020).

We choose a point in the middle of the 2019 grid, close to where the measurement site of the Uppsala University is located,
see the black cross Figure 1b. The water table height at this location is plotted together with the meltwater input from the EBFM for the whole run and for the last years in Figure 6. It can clearly be seen that the aquifer reacts almost instantly to a meltwater input peak. There is a time delay between a peak in meltwater input and aquifer water table height in the order of weeks. This delay is likely caused because this location is in the centre of a topographic low. It can therefore receive water from all sides, which extends the growing phase of the aquifer in this location beyond the initial pulse. The strongest meltwater
peaks coincide with sharp water table height increases. Once the water table depth peak has passed, the water table decreases again with a rate dependent on the peak magnitude but following the same smooth pattern every year.

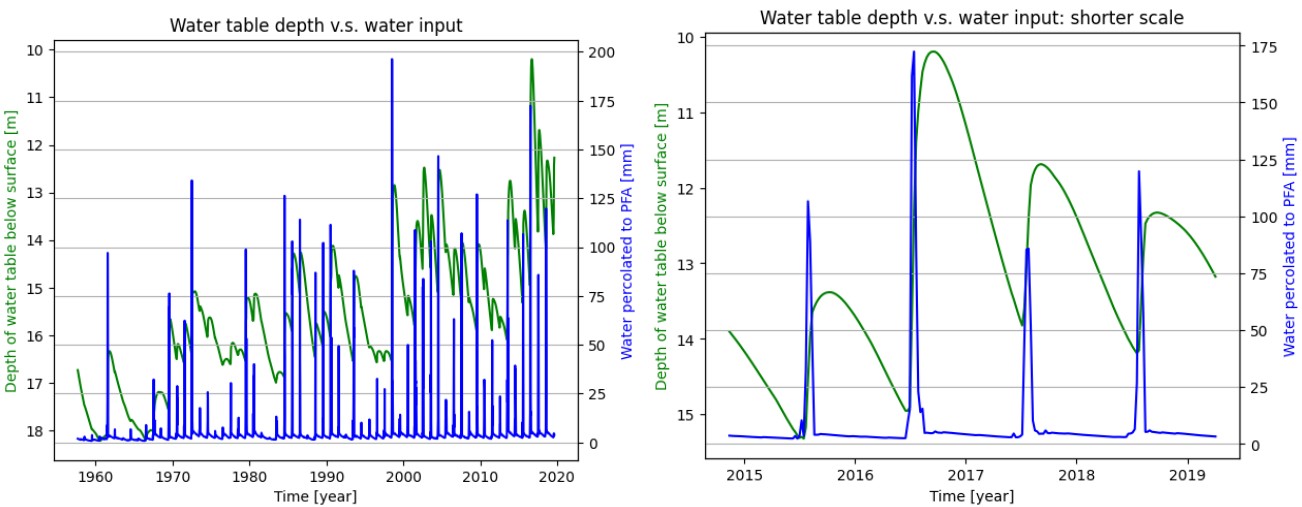

**Figure 6.** Water table depth from LPFAM versus meltwater input from EBFM for the whole simulation (left) and for the final years (right).




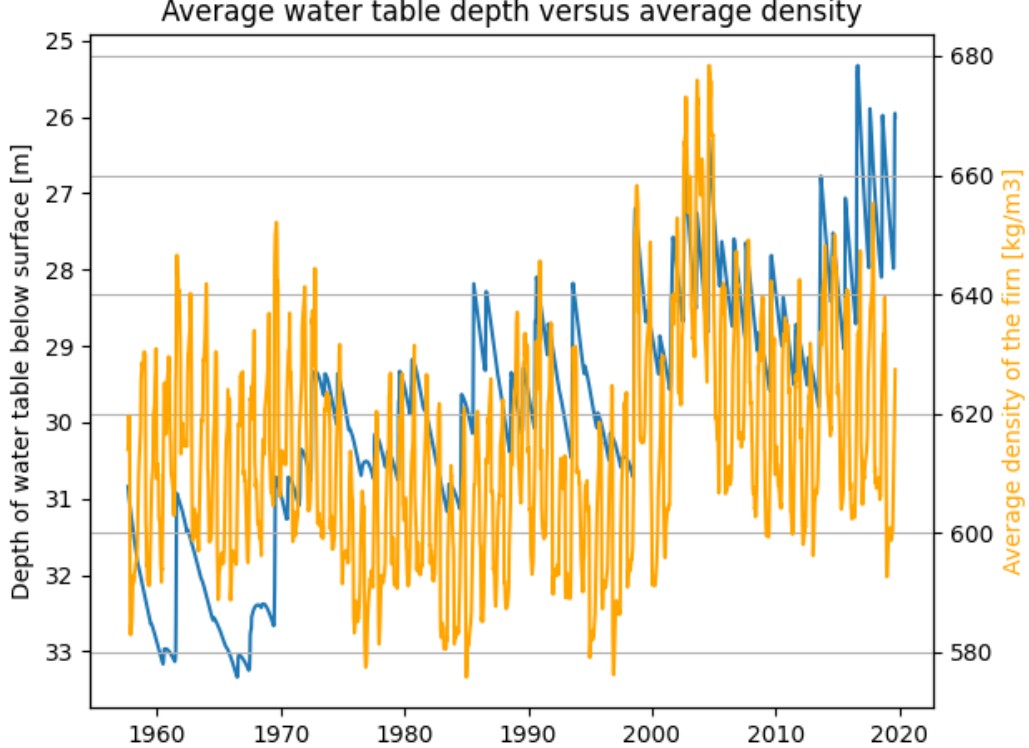

**Fig 7** Total water content of the aquifer (orange) with the average water table height (left). Average density on the modelled grid from the EBFM (orange) (right).

In Fig 7, the average density on the model grid versus the average water table depth is shown. The R-correlation between water table depth and average density is -0.39 (negative, as a higher density corresponds to a lower water table depth). There are two time-varying input files provided to the aquifer model: density and meltwater input. The weak correlation between average water table depth and the average density tells that the variance in the water table depth is determined by the variable meltwater input, and only partly by a varying density.

## 4.4 Uncertainties

There are processes missing in the LPFAM, that might change the results or become the focus of further research. A main source of uncertainty is that there is only a one-way coupling from EBFM to LPFAM. Two-way coupling would be more physically desirable. For example, the extra weight of water in saturated firn will induce faster densification by the gravitational overburden pressure, a process currently not incorporated in the EBFM. Furthermore, as soon as the aquifer reaches closer to the surface, the overlaying firn may not be able to protect the water from refreezing. The observed water table does not get closer than 10 metres below the surface, so for the present-day conditions this is unlikely to happen. However, in the future,





with less pore space and more meltwater production refreezing might occur. If refreezing happens, an ice lens might form. This ice lens then entirely blocks the water table from rising, or meltwater from the surface to reach the aquifer, effectively creating a small confined water storage unit. The PFA is, if that occurs, not entirely unconfined. A two-way coupled EBFM -
LPFAM, where on top of the current couplings, refreezing of PFA water and densification of saturated firn are accounted for, could be the scope for further research.

Another source of modelling errors is water drainage through crevasses, which is not accounted for. Instead, all water drains through the lateral boundaries of the grid. No data is available on the existence, location, size and capacity of the sinks within our modelled grid on Lomonosovfonna, so they could not be incorporated to the LPFAM. The only sink for water in the
LPFAM is horizontal movement through the boundaries of the model domain. As can be seen in the results for 2017, the LPFAM overestimates the water table height in the south-eastern corner of the model grid. There is likely a crevasse field here, but we do not know the specific locations and dimensions of the crevasses. Also, the interaction between the aquifer and crevasses, in particular the drainage rates, are unknown. Preliminary tests with artificial drains in this region improved the results. Identification of crevasse fields in satellite products, e.g. Sentinel-2, is challenging given high snow accumulation
rates, but could provide important data on PFA sinks in the area.

In a future climate, increased melting and rainfall (Hansen et al., 2014, Bintanja and Andry, 2017) will lead to less deep and higher density firn, and will lead to a water table depth that reaches closer to the surface. This will simultaneously increase the likelihood for refreezing of the water table in places where the water table is close enough to the surface (<8-10 m). During the modelled period (1957-2019) this has not occurred yet, as water table depth did not reach closer than 1.3 m below the
surface, but for the future development of the PFA, this could become important, as well as the formation of surface streams.

The hydraulic conductivity we find by tuning the LPFAM is in between the values of $1 * 10^{-3} - 1 * 10^{-5}$ [m s$^{-1}$] (Fountain and Walder, 1998, Miller et al., 2017, Stevens et al., 2018, Miller et al., 2020, Miller et al., 2023). We find this uniform hydraulic conductivity by manually tuning our model until it fits with observations. A tuned variable can only represent a real parameter
if all other process at play are perfectly represented, so our tuned variable can compensate for errors in the firn conditions from the EBFM, uncertainties in observations and/or missing processes.

## 5. Conclusions

In this study, we adapted an existing groundwater flow model MODFLOW6 so that it can be applied to model a perennial firn aquifer. We test the model setup on a PFA, that has been observed on the Lomonosovfonna ice cap in central Svalbard. We use in situ measurement campaigns of three consecutive years to tune uncertain parameters, so that the modelled water table height fits with observations. From the calibration we infer a hydraulic conductivity of the PFA of $6.4 * 10^{-4}$ m s$^{-1}$. We run the PFA model from 1957 – 2019 and find a steady increase in water table height over the modelled period. We find that the PFA likely already existed before 1957. The height of the water table reacts quickly to meltwater input, in the order of weeks, and long-term water table depth variations further depend on long-term firn density changes. During the cold season, the water table decreases steadily to a minimum prior to the new melt season. The water table is likely to rise further in the future, as more melt/rain and a denser firn pack are expected in a warmer climate. This may newly induce refreezing of the PFA from above. With continued firn densification, the water table is likely to locally reach the surface in the coming decades.

**Data and code availability**

Datasets used as input: the NPI Digital elevation model from (Melvær et al., 2014), density and meltwater input from the EBFM (Van Pelt et al., 2019). MODLFOW 6 and the adapted python package to generate the configuration files are open-source and freely available, see Bakker et al. (2016), Langevin et al. (2017). LPFAM model data, code and GPR observations available on request.

**Author contributions**

TvdA designed the LPFAM and tuned it, and performed the main simulation. WvP provided the EBFM input files, and provided feedback and support during the LPFAM modelling. RP refined the GPR data into usable water table depth datasets. VP initiated and coordinates the field work expeditions. TvdA prepared the manuscript, with contributions from all authors.

**Acknowledgements**

TvdA received funding from the NPP programme of the NWO. WvP acknowledges funding for the field measurements on Lomonosovfonna from Svalbard Integrated Arctic Earth Observing System (SIOS), Stiftelsen Ymer-80 and Finn Malmgrens stipendiestiftelse. VP acknowledges a grant from The Swedish Science Council supporting the observations on Lomonosovfonna.



The authors declare no competing interests.

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
