# Peer review of "Long-term development of a perennial firn aquifer on the Lomonosovfonna ice cap, Svalbard"

_EGUsphere, 2024_

## Referee Comment (RC2)

[referee-annotated manuscript omitted]

---

## Author Comment (AC1)

The subject matter of this paper is highly relevant and worthy of investigation as it concerns understanding the future mass balance of glaciers, ice caps and ice sheets globally. The datasets used in this study, which include a combination of valuable field data (GPR and dGPS), and state-of-the-art firn densification, and flow models should be sufficient to address the main objectives of this study. Scientific rigor however is absent in several areas, particularly in supporting primary conclusions of this paper. For example, one of the main results, ie. PFA has persisted since the 1950's, is based on reanalysis data, for which there has been no attempt to establish quantitative relevancy to the study site. There are several instances where correlations between datasets, which are not present, would be helpful to provide more informed statements about the conclusions of this paper. Also, there has been very little attempt to explore other evidence to support the primary conclusions, as discussed below.

This paper is fraught with poor writing quality, numerous errors in punctuation, inconsistent formatting, and very poor sentence structure throughout most of the paper. Some sections however are better than others. Also, there are many problems with the figures (discussed below) which make interpretation of the results of this paper challenging. As a result, a thorough assessment of the scientific validity of this paper is not possible.

We thank the reviewer for a thorough analysis of our paper and agree that the conclusion on the PFA existence in 1957 could be made clearer. We argue that the Nordli et al (2020) timeseries are a useful proxy for the general climate of the Svalbard archipelago during the years before our modelling. We argue that this information is useful to draw conclusions on aquifer presence in 1957, as discussed in more detail in our reply to the first specific comment. We further highlight that the meteorological forcing dataset (NORA10) is elevation dependent and has previously shown good performance for temperature against AWS data at various elevations and on different glaciers in Svalbard (see Van Pelt et al. 2019 for more details). Finally, we agree that the writing and figures needed improvement and will make thorough revisions throughout the manuscript.

**Specific Comments** (individual scientific questions/issues)

L18 (abstract): The statement " *We find that the aquifer was present in 1957, and…*" is unsubstantiated based on the evidence provided. Specifically, the main point of evidence given within the paper for the PFA to be present in 1957, ie. "*Given that temperatures during 1957-1977 were likely cooler than during 1937-1957 (Nordli et al., 2020)…*" is not even a certainty so how can this 'finding' that the PFA existed in the 1950's be stated with absolute certainty, as it is in the abstract? Further more, the temperature data used in the model, ie. Riestad, (2011), is downscaled to ocean surface. At the very least, a correlation between the longer term Riestad data (downscaled over ocean surfaces) and the data collected from the Upsalla University AWS (~1000 m a.s.l.), should be stated.

Also, why is Nordii used as the reference for existence of the PFA, where the Riestad data was used to drive the model? Beyond the model results, this paper does not explore any further evidence to support, or question existence the PFA 'existed' in the

1950's. Is there no other evidence to support, or refute the possibility that preferential water flow routing may have been affected by changes, in ice dynamics, topography, thickness, and/or firn density over this 60+ year period of time? At the very least, it would be helpful to include a figure of near-surface air temperatures from the nearest station in order to provide the reader with some sense of the magnitude of temperature variability over the 60+ year period of this study. Simple calculations of the depth of penetration of the cold wave, and the modeled temperature regime that existed at depths to the bottom of the PFA.

We agree with the reviewer that our line of reasoning supporting the notion that the perennial firn aquifer existed before 1957 should be made clearer. The timeseries in Nordli et al. (2020) show homogenised observational timeseries for Svalbard (Longyearbyen) back to 1898. The Reistad et al. (2011) dataset only goes back to 1957. Fig. 3 in Nordli et al. (2020) reveals that the decades before 1957 (~1930-57) were warmer than the period 1957-1977. Even though these timeseries apply to sea level we argue it is unlikely that an opposite temperature trend would apply at high elevations. Since we used the colder period 1957-1977 to initialize our model, the presence of a perennial firn aquifer at the end of the spin-up period is a strong indication that a perennial firn aquifer would also have been present in our model in 1957 if we would have been able to initialize the model for the actual (warmer) period 1937-1957. We argue that higher temperatures imply more melt and growth of the firn aquifer as this is also seen during warm years throughout our simulation period. Based on the likely warmer temperatures prior to 1957 than the period that was used for model spin-up, we could rather argue that the perennial firn aquifer was larger than the size it currently has at the start of the long run in 1957. Hence, we argue that our conclusion about likely firn aquifer presence is justified, and may in fact be rather conservative. We will rewrite L296 – 298 according to the detailed comment below.

The Reistad et al. (2011) dataset was used to drive the EBFM to generate, among others, firn characteristics, as described in Van Pelt et al (2019). The NORA10 HIRLAM regional climate model based reanalysis dataset from Reistad et al. (2011) is originally available at 10-km spatial resolution and is already elevation dependent (topography is accounted for in the regional climate modelling). These data hence do not apply to sea level. In Van Pelt et al. (2019) it is described how this dataset is further downscaled to a 1-km spatial resolution and further corrections are made there to project air temperature and precipitation onto the finer-scale elevation grid. For more details on this downscaling procedure we refer to Section 2.1 in Van Pelt et al. (2019). Van Pelt et al. (2019) also compared downscaled air temperature timeseries against AWS observations on various glaciers/ice caps in Svalbard (Kongsvegen, Holtedahlfonna, Austfonna and Nordenskiöldbreen), showing good performance (see Table 3 in Van Pelt et al. 2019). For example, the nearest AWS on Nordenskiöldbreen at 520 m a.s.l. (only about 10 km's from the PFA area) shows an R-correlation of 0.95 and a bias of 0.8°C for 3-hourly temperature data between 2009 and 2015. In general, the comparison with the AWS data, covering an elevation range of 0-700 m a.s.l.) across Svalbard does not indicate that there are elevation-dependent biases in the temperature forcing dataset.

We agree with the reviewer that there could well be other factors controlling the development of the perennial firn aquifer and the way water moves through the firn. Changes in ice dynamics, topography and ice thickness are not taken into account in this study, changes in firn thickness and density are however included in our modelling. Regarding the ice dynamics, the Lomonosovfonna perennial firn aquifer is located on a very slow moving ice cap, close to the ice divide between several glaciers. The surface height change rates at our study site have been close to zero during our modelling period, see Fig. 2 in Geyman et al (2022). Hence, there are no indications that the ice geometry and flow speeds (e.g. affecting crevassing) have markedly changed throughout the simulation period.

L152 "*Furthermore, fast deep percolation is modelled using the parameterization by Marchenko et al. (2017)*". Question: Is percolation of meltwater not being modelled in 2 ways then, ie. by the Van Pelt EBFM AND the Marchenko model ?

Yes, it is indeed. 'Slow' percolation is modelled with the tipping-bucket method , fast percolation through preferential pathways is modelled by using the parameterization proposed by Marchenko et al (2017). This combined model is described by Marchenko et al. (2017) and implemented in EBFM (Van Pelt et al. 2019). We will change the text (L147 – L154) to make it more clear:

'Water in the EBFM originates from surface melt and rain, and percolates down from the surface into the firn. This happens in two ways: by using the  fast deep percolation statistical parameterization from Marchenko et al. (2017), and by applying the 'tipping bucket' scheme on the remainder. The fast percolation instantaneously distributes water in the vertical model according to a normal distribution with a peak at the surface. Further water transport is modelled with a tipping-bucket approach. First, the water refreezes when the conditions in a model layer are sufficient, being that the temperature is below melting point and the density below the density of ice. Refreezing raises the temperature and density. If not all water refreezes, a small portion will be stored in the layer as irreducible water content. The remaining water will percolate down to the next layer, where the process repeats. This continues until the water encounters impermeable ice, in which case the water becomes runoff (i.e. water input in the LPFAM).'

L296-298:
 "*At the end of our spin-up routine in 1977 a PFA is present in the firn. Given that temperatures during 1957-1977 were likely cooler than during 1937-1957 (Nordli et al., 2020), we argue it is highly likely that a PFA was already present in 1957 and in the preceding decades.*"
This argument that a PFA was already present in 1957, is not convincing as it is poorly supported with scientific proof, or even a convincing argument.

We will rewrite L296 – 298 according to the answer to the first specific comment above:

' The timeseries in Nordli et al. (2020) show homogenised observational timeseries for Svalbard (Longyearbyen) back to 1898. Fig. 3 in Nordli et al. (2020) reveals that the decades before 1957 (~1930-57) were warmer than the period 1957-1977. Even though these timeseries apply to sea level we argue it is unlikely that an opposite temperature trend applies at high elevations. Since we used the colder period 1957-1977 to initialize our model, the presence of a perennial firn aquifer at the end of the spin-up period is a strong indication that a perennial firn aquifer would also have been present in our model in 1957 if we would have been able to initialize the model for the actual (warmer) period 1937-1957. We argue that higher temperatures imply more melt and growth of the firn aquifer as this is also seen during warm years throughout our simulation period. Furthermore, high melt rates were also listed as a requirement for firn aquifer formation in Kuipers Munneke et al. (2014). Based on the likely warmer temperatures prior to 1957 than the period that was used for model spin-up, we argue that the perennial firn aquifer may have been larger than the size it currently has at the start of the simulation in 1957.'

L120: this statement ... '*Firn density and meltwater input to the aquifer is required to model a PFA.*' comes across as a general statement about modeling a PFA. I'm sure this is not what is intended by the author as it ignores many other factors that could be relevant. Please clarify.

That is correct. We do not want to imply that firn density and meltwater input are the only fields neccesary to model a PFA, far from it. We want to highlight here that for our approach, by using a groundwater flow model, we need as input information about the firn pack, in particular firn density and meltwater percolation. One can obtain those from observations, built a firn model within a PFA model or take those two fields as output from an existing firn model.

We will rewrite this to:
'In this model study, time-dependent firn density and meltwater input are the primary input variables used to model the PFA found at the Lomonosovfonna ice cap.'

L255-260:  re: 2017 non-coherent data.... Could this not have been simulated by deriving the correlation values for the areas in common over the 3 time periods, ie remove all points from  areas in 2017 that do not exist in the 2018 and 2019 datasets. This may provide some quantitative basis to refute or support the reason for coherence in the 2017 data.
Also, could there have been other factors responsible for the low coherence in the 2017 data, eg, towing speed of radar?, rough surface topography?

We thank the reviewer for this useful suggestion. We checked, and found that in 2017, only 11% of all observational points of that campaign lay within the 2018 and 2019 grid, which is about 3800 datapoints. If we would restrict our data to just the 2018 and 2019 grid, we would therefore lose almost 90% of the datapoints collected in 2017, which is undesirable.

Yes, there could have been other factors that influence the non-coherence. Missing effects from not including topography changes and ice dynamic effects can influence PFA formation and water movement. However, as stated in the answer to the first specific comment, our observational area has not experienced major geometric changes (Geyman et al. 2022) and ice flow is slow due to the proximity to an ice divide.

Furthermore, the source of non-coherence from towing speed and rough, possibly slightly different surface topography are to a first order represented by the difference in 2019 and 2018 data. Our observations show only slight differences, so we are confident in stating that the effect of differences in observational conditions (such as towing speed and slightly different topography) are small, and not enough the explain the error between modelled and observed water table depth in 2017, which is in the order of meters.

**Technical Corrections**

Fig 1: - Significant place names should be added to the left hand side figure. Include label location of the Svalbard airport (as it is mentioned in the text). The figure caption is poorly worded and hard to understand.

We will add Longyearbyen Airport, and rewrite the figure caption.

I don't see 2015 or 2016 on the figure, only in the caption

We removed the years 2015 and 2016 because of their sparsity in useable data points, we will remove it from the caption

- 2017 looks green on the fig, indicated as yellow in the caption

Correct, a remnant of the deleted years from the previous comments. We will change this accordingly

- In sentence 2, its not clear what 'red rectangle' you are referring to – left or right hand figure.

Left hand side, we will change the color and the sentence to make it easier to distinguish.

For this sentence...."The rectangles correspond to the minimum and maximum coordinates of the measurement in those years and thus show the extend of the PFA measurements" .
- 'Extend' should be 'extent' (2 instances), turned should be converted,

We will change this

- Avoid over-use of first-person pronouns - it is very distracting to the readability of the paper. This is particularly an issue in the results section 4.1 and conclusion.

We will reduce the amount of first-person pronouns in the paper

- Please correct citations to use semi-colon, not comma to separate a list of references. An example of this is on line 50, but it happens throughout.

We will do this

- Please follow the rules of the journal when referring to figures and tables, noting the differences between use of these words at the beginning of the sentence versus in running text. Many errors of this kind throughout the paper.

We will read the guidelines of the journal and adjust the text accordingly

- Please note that the word "Table" is never abbreviated and **should be capitalized when followed by a number (e.g. Table 4).**

We will change this

- You have 'in situ' and 'in-situ' throughout, please be consistent with formatting rules.

We will use 'in situ'

L165: remove the word 'in' from the bracketed text.

We will do this

L215: 'transfer' doesn't seem like the right word... 'transform' or 'convert' perhaps?
    -remove space before comma

We will use 'convert' and remove the space before the comma

L238:  Improper use of brackets

We will remove 'e.g.' and write the list of studies as a 'normal' end-of-the-sentence reference. Also, we will change the comma to a semicolon

L216: change 'will' to 'is'
- It is not common for a DEM to have a variable cell size - please clarify.

This is a mistake on our side, the DEM we use has a constant cell size, which is 5 meters. We interpolate it to our grid, which has rectangular cells (72 x 96 meters) to accommodate 100 cells in every direction while maintaining a rectangular area around our observation locations. We will add this to the text.

L218: the differences here needs to be stated. please indicate rmsd between the datasets

We will rewrite this, this is from an older version of the model. We do not use the topography observations anymore: we simply use the depth to the water table directly from the GPR, and use the DEM from Melvaer et al (2014) to turn LPFAM output from height above sea level to depth to the water table.

L235: close off  should be close-off

We will change close off to close-off throughout the manuscript

L262: no hyphen in northwestern or southwestern

We will change this throughout the text

L263: include the max and min differences between the modelled and observed water table depths.

We will add this

L310; this sentence makes no sense on its own.

We are not sure to what sentence the reviewer is referring. The two sentences can be shortened and combined to: ' The water table depth and meltwater input at a single modelled grid cell, where the measurement station is located, is shown in Figure .... '

L325: should be Fig. not Fig

We will change this

L337: "*If refreezing happens, an ice lens might form*." This is a weird statement -  Why would ice not form if refreezing 'happens'?

We will change this to 'Meltwater refreezing creates an impermeable ice layer in the firn'

L349: improve the results by how much? Enough to explain the absence of crevasses as being the reason why coherence was so much lower?

We will provide the RMSE, and add a figure to the supplementary material in which we show the location of the added crevasses, and a discussion on the additional assumptions made.

L220-240: this section is misplaced as it describes methods, not results.

We agree and will move this part to the methods section

L252:  Change...

'*The modelled water table is overestimated in the south-eastern corner*'
 to
'*The modelled water table **depth** is overestimated in the south-eastern corner*'.

We will change this

L228: consequence is misspelt.

Thanks for noticing, we will change this

L320 and beyond, please refer to 'density' as 'firn density'

We will add this thourghout the manuscript.

Figure 2: y-axis – specify 'above mean sea-level'

We will change this

Figure 4. it would be far easier to compare the spatial pattern of differences between observed and modelled if the same color scheme for water table depth was used for both. Or, a third plot could be added to illustrate the differences.

We note that Figure 4 shows on the left side the modelled water table depth with the observation locations, and on the right already the difference between modelled water table depth and the observations. We see that the title of the right plots is misleading, it should be modelled water table depth difference wrt observations [year]. We will change this

It would be very useful here to have a box superimposed on the 2017 results which indicates the extent/positioning of the 2018/19 results.
Great suggestion, we will add this
- Caption refers to 'left column' and 'right'.
We will add 'right column'

Figure 5: units missing on both x and y axis.
We will add [m]
Figure 7:
- Fix caption (ie., Fig 7 to Figure 7.)
We will change this
- Color code left hand axis title to match blue line.
We will do that
- Refer to water table line as blue.
We will do this
- Not sure you need to specify both 'right' and 'orange'
We will only refer to 'orange' and 'blue'

---

## Author Comment (AC2)

**Comments by reviewer 2**
The original comments were in a PDF and have been copied here (in black). Answers are provided in blue.

Line 56-58: How is this model different from the model being used in this manuscript? Since this is the most recent model, please point out the detailed differences including assumptions and pros and cons of each approach.

Advantages include the combination of thermodynamics and firn modelling with the aquifer modelling in Miller et al (2023), and includes procedures to model unsaturated and saturated flow, a process that is missing in our approach. SUTRA-Ice is only tested on a 2D, flowline case of the Helheim glacier with constant recharge rates. Our simulation is 3D, tested and calibrated against three years of observations and including a 'downscaled' meteorological forcing to simulate surface melt, firn conditions and the behaviour of the aquifer over time.

We will add to L 57: 'The model SUTRA-ICE is comprehensive: it contains water flow through the unsaturated zone as well as water movement within the saturated zone. Freeze-thaw cycles are modelled, so a winter freeze of (a part of) the modelled PFA is represented. The model is tested on a 2D flowline of the Helheim glacier, with constant recharge rates. 3D flow and realistic meltwater input from a (downscaled) climate or energy balance model is missing.'

Fig 1: add ice edge boundary to this figure so the reader has a better sense of glacier setting. Also, make sure the colors of the rectangles match the caption and the years 2015 and 2016 do not appear in the text. Change 'next paragraph' to 'below'

We will remove the years 2015 and 2016, both from the figure as from the caption. We will change the colormap to make it easier to distinguish between height contours. We do not deem it necessary to add glacier outlines to this graph, because in this study we do not refer to or use data from individual glaciers.

Line 111: Explain this in more detail. Are there any density vs depth measurements available?

Yes, snow pits have been dug at this location, to which the EBFM is tuned, see Van Pelt et al. (2019). We then use a density profile of the EBFM close to the location of the measurements to calculate the dielectric constant for firn snow, which we use to calculate the velocity of the radar wave in the firn. We then use this velocity of the radar wave to turn the TWTT to the reflective surface that is semi-automatically picked to a depth to the water table. We will change Ln 111 – 114 to:

'The raw GPR data was minimally processed with zero-time adjustment and a low-pass filter (300 MHz cut-off frequency). An example radargram is shown in Figure 2. The water table is picked from a radargram shown in Figure 2 semi-automatically: first, the reflective surface of the water table is manually found in a single data point. Then, a tracing algorithm is used to track that reflective surface through adjacent points. This

results in the two-way travel times (TWTT) per datapoint. Then, the velocity of the radar wave in the firn is used to calculate the distance to the water table from the surface. For this, the dielectric constant of firn is required, which is calculated according to Eq 1 from Kovacs et al (1995) where $\rho f$ is the density of the firn layer and $\rho w$ the density of water:

The density of the firn at the location of the observations is obtained from the Energy Balance Firn Model (EBFM; Van Pelt et al, 2019), of which a description is given in the next section.'

Line 123: please give more details

We will rephrase this to:
' … which has previously been calibrated and validated against stake measurements, weather station data and observed density profiles from shallow firn cores. For more details, we refer to Van Pelt et al (2019).'

Line 140: is there a term that describes the density changes due to melt/freeze events?

Yes, the second F/dZ term in Eq 4. You can choose what unit for F to use, in the manuscript we used kg m$^{-2}$ s$^{-1}$ , so we needed to divide by the layer thickness dZ. We will just use F in kg m$^{-3}$ s$^{-1}$ as the density increase through refreezing.
We will add to Line 140: and F is the refreezing rate (in kg m$^{-3}$ s$^{-1}$) and remove dZ from the equation.

Line 219: can you quantify the difference?

Thanks for this suggestion. However, we rather do not use the GPS measurements of the surface height. We will leave the GPR observations in terms of 'depth below the surface', and transform LPFAM data, which is typically in height above sea level with the DEM from Melvaer et al (2014) to a depth below the surface. We will change Ln 211 – 2019 to:

'LPFAM output gives the elevation of the water table in meters above sea level (m a.s.l.). The raw observational data gives instead water table depth below the surface. We use the Digital Elevation Model (DEM) of Svalbard referred to as the Terrengmodel S0 with a resolution of 5 meters from ,Melvær et al. (2014) re-gridded to the model grid of the LPFAM as the firn surface. We then use the re-gridded DEM to subtract our modelled water table height above sea level, to obtain modelled water table depths in m a.s.l.'

Table 2: Can you estimate the uncertainty in this?

Thanks for this suggestion.  We will add to Line 292:

'There are uncertainties in the observed water table depth shown in this study. The system specific uncertainty (related to the sampling frequency, the cable length, and

the GPR used is small and about ±0.02 meters. The largest source of uncertainty stems from the calculation of the velocity of the radar wave, which is calculated using a modelled density profile of the firn. When changing the firn density arbitrarily with ±10%, this resulted in a spread of ±0.21 m in calculated water table depths. The uncertainty arising from digitizing the water table, quantified by doing cross analysis of double-measured points during the same field season, results in ±0.03 m.

Fig 7: should this be blue?

Yes, thanks for spotting. We will change this.

Ln 354: Sentiel-1 may also be helpful and could potentially identify buried crevasses since C-band SAR can penetrate some snow covered surfaces

This is a great suggestion. We will add Sentinel 1 to our discussion and look for more prove on the existence of crevasses.

Ln 355: or potentially lakes depending upon local topography

We will add 'or meltwater lakes, if the surface topography allows for it'

Melvær, Y., Aas, H., and Skoglund, A.: Terrengmodell Svalbard (S0 Terrengmodell), Norwegian Polar Institute, 465, 2014.

---

## Author Response (AR1)

The subject matter of this paper is highly relevant and worthy of investigation as it concerns understanding the future mass balance of glaciers, ice caps and ice sheets globally. The datasets used in this study, which include a combination of valuable field data (GPR and dGPS), and state-of-the-art firn densification, and flow models should be sufficient to address the main objectives of this study. Scientific rigor however is absent in several areas, particularly in supporting primary conclusions of this paper. For example, one of the main results, ie. PFA has persisted since the 1950's, is based on reanalysis data, for which there has been no attempt to establish quantitative relevancy to the study site. There are several instances where correlations between datasets, which are not present, would be helpful to provide more informed statements about the conclusions of this paper. Also, there has been very little attempt to explore other evidence to support the primary conclusions, as discussed below.

This paper is fraught with poor writing quality, numerous errors in punctuation, inconsistent formatting, and very poor sentence structure throughout most of the paper. Some sections however are better than others. Also, there are many problems with the figures (discussed below) which make interpretation of the results of this paper challenging. As a result, a thorough assessment of the scientific validity of this paper is not possible.

We thank the reviewer for a throurough analysis of our paper and agree that the conclusion on the PFA existence before 1950 could be made clearer. Related to the temperature dataset used, we use the long-term temperature dataset from Nordli et al (2020) as a proxy for the general climate of the Svalbard archipelago during the years before our modelling, measurements and reanalayis data is available. We believe that this dataset offers sufficient insight in the temperature regime (note that we did not use the numerical values of the temperatures of Nordli et al (2020) in neither the EBFM nor the LPFAM). Detailed plans on how to improve this will be given as answers to the specific questions below. We will thouroughly reread the paper and improve the writing and the figures.

**Specific Comments** (individual scientific questions/issues)

L18 (abstract): The statement " *We find that the aquifer was present in 1957, and…*" is unsubstantiated based on the evidence provided. Specifically, the main point of evidence given within the paper for the PFA to be present in 1957, ie. "*Given that temperatures during 1957-1977 were likely cooler than during 1937-1957 (Nordli et al., 2020)…*" is not even a certainty so how can this 'finding' that the PFA existed in the 1950's be stated with absolute certainty, as it is in the abstract? Further more, the temperature data used in the model, ie. Riestad, (2011), is downscaled to ocean surface. At the very least, a correlation between the longer term Riestad data (downscaled over ocean surfaces) and the data collected from the Upsalla University AWS (~1000 m a.s.l.), should be stated.

Also, why is Nordii used as the reference for existence of the PFA, where the Riestad data was used to drive the model? Beyond the model results, this paper does not explore any further evidence to support, or question existence the PFA 'existed' in the 1950's. Is there no other evidence to support, or refute the possibility that preferential

water flow routing may have been affected by changes, in ice dynamics, topography, thickness, and/or firn density over this 60+ year period of time? At the very least, it would be helpful to include a figure of near-surface air temperatures from the nearest station in order to provide the reader with some sense of the magnitude of temperature variability over the 60+ year period of this study. Simple calculations of the depth of penetration of the cold wave, and the modeled temperature regime that existed at depths to the bottom of the PFA.

We agree with the reviewer that our line of reasoning supporting the notion that the perennial firn aquifer existed before 1957 should be made clearer. Temperatures were not 'likely' higher before 1957 than during the period, they were actually observed to be higher at Longyearbyen airport. Since the Lomonosovfonna is in relatively close proximity to the location of the observation (only higher above sea level) we conclude from the dataset from Nordli et al (2020) that it must have been warmer at the Lomonosovfonna ice cap during the period before 1957 than during the period 1957 – 1977 as well. Since the perennial firn aquifer develops during our spinup (which is done over the period 1957 – 1977) and grows during our simulation from 1957 – 2019, we conclude that conditions were feasible for a perennial firn aquifer to sustain and grow during the relatively cold period 1957 – 1977. Higher temperatures will mainly make it easier for the perennial firn aquifer to survive because of more meltwater input from the surface and less chance of refreezing in the firn pack. We will rewrite L296 – 298 according to the detailed comment below.

The data from Reistad et al (2011) is not used in the model presented in this paper. It is used to drive the EBFM to generate firn characteristics, which is also published in Van Pelt et al (2019). We refer the reviewer to that paper for a more extensive discussion of how the elevation-corrected data from Reistad et al (2011) is used. As the EBFM and the tuning of it is not a topic of this paper, but extensively discussed in Van Pelt et al (2019), we do not deem it nessecary to add correlations between the data of Reistad et al (2011) and the data obtained by the Uppsala University.

Regarding the choice for Nordli et al (2020) over Reistad et al (2011) to support the argumentation on why the perennial aquifer likely existed before 1957: the dataset of Reistad et al (2011) does not extend to before 1957.

We agree with the reviewers that there could well be other factors controlling the development of the perennial firn aquifer and the way water moves through the firn. Changes in ice dynamics, topography and ice thickness are not taken into account in this study, changes in firn thickness and density are part of the LPFAM. Regarding the ice dynamics, the Lomonosovfonna perennial firn aquifer is located on a very slow moving ice cap, close to the ice divide between several glaciers. The mass change rates at our study site have been close to zero during our modelled period, see Geyman et al (2022), so changes in ice thickness, ice dynamics and the topography we deem small and insignificant at our study area.

L152 "*Furthermore, fast deep percolation is modelled using the parameterization by Marchenko et al. (2017)*". Question: Is percolation of meltwater not being modelled in 2 ways then, ie. by the Van Pelt EBFM AND the Marchenko model ?

Yes, it is indeed. 'Slow' percolation is modelled with the refreezing method from the EBFM, fast percolation through preferential pathways is modelled by using the parameterization proposed by Marchenko et al (2017) but built into the EBFM. We will change the text to make it more clear. We will change L147 – L154 to:

'Water in the EBFM originates from surface melt and rain, and percolates down from the surface into the firn. This happens in two ways: by using the fast deep percolation statistical parameterization from Marchenko et al. (2017), and by applying the 'tipping bucket' scheme on the remainder. The tipping bucket is implemented as follows: First, the water refreezes when the conditions in a model layer are sufficient, being that the temperature should be below the melting point and the density should be lower than the density of ice. Refreezing raises the temperature and density. If not all water refreezes, a small portion will be stored in the layer as irreducible water content. The remaining water will percolate down to the next layer, where the process repeats. This continues until the water encounters a layer that has the density of ice, where it will pile up to fill the pore spaces of the firn above the ice. The percolated meltwater as function of the depth is then the sum of what was transported by both methods.

L296-298:
 "*At the end of our spin-up routine in 1977 a PFA is present in the firn. Given that temperatures during 1957-1977 were likely cooler than during 1937-1957 (Nordli et al., 2020), we argue it is highly likely that a PFA was already present in 1957 and in the preceding decades.*"
This argument that a PFA was already present in 1957, is not convincing as it is poorly supported with scientific proof, or even a convincing argument.

We will rewrite L296 – 298 according to the answer to the major comment above:

At the end of our spin-up routine in 1977 a PFA is present in the firn. In other words, the firn- and climatic conditions during 1957 – 1977 are favourable for PFA formation. Temperature is a strong indicator for the the existence of PFA's, see for example Kuipers Munneke et al (2014), as higher temperatures make refreezing of the PFA less likely during the winter and lead to more meltwater input into the firn. More meltwater input into the firn will (obivously) foster the growth of a PFA, but will also increase firn temperatures because it exchanges (latent) heat with its surroundings. This will make

the firn temperate, an ideal condition for a PFA to persist. In Figure 7 in  Van Pelt et al (2019) it can be observed that the firn is temperate in many places in Svalbard, and that the trend over the period 1957 – 2017 was almost zero at the Lomonosovfonna ice cap. If the PFA can form and sustain, and the firn stayed temperate during the cold 1957 – 1977 period, we deem it highly likely that the PFA already existed before 1957.

L120: this statement … '*Firn density and meltwater input to the aquifer is required to model a PFA.*' comes across as a general statement about modeling a PFA. I'm sure this is not what is intended by the author as it ignores many other factors that could be relevant. Please clarify.

That is correct. We do not want to imply that firn density and meltwater input are the only fields neccesary to model a PFA, far from it. We want to highlight here that for our approach, by using a groundwater flow model, we need as input information about the firn pack, in particular firn density and meltwater percolation. One can obtain those from observations, built a firn model within the PFA model or take those two fields as output from an existing firn model.

We will rewrite this to: In this model study, time-variant firn density and meltwater input are the primarily input variables used  to model the PFA found at the Lomonosovfonna ice cap.

L255-260:  re: 2017 non-coherent data…. Could this not have been simulated by deriving the correlation values for the areas in common over the 3 time periods, ie remove all points from  areas in 2017 that do not exist in the 2018 and 2019 datasets. This may provide some quantitative basis to refute or support the reason for coherence in the 2017 data.
Also, could there have been other factors responsible for the low coherence in the 2017 data, eg, towing speed of radar?, rough surface topography?

We thank the reviewer for this usefull suggestion. We checked, and found that in the common area where all three years observations where made, there is a 100% match in locations between 2018 and 2019. That means that all points of the year 2019 are in the same 'box' as the year 2018 (note that this does not mean that all points are also made at exactly the same locations. In both years, the amount of observation locations is about 20000 datapoints.

In 2017, only 11% of all obsvervational points of that campaign lay within the 2018 and 2019 grid, which is about 3800 datapoints. If we would restrict our data to just the 2018 and 2019 grid, we would therefore lose almost 90% of the datapoints collected in 2017. This is something we would not like to do.

Yes, there could have been other factors that influence the non-coherence. Missing effects from not including topography changes and ice dynamic effects can influence PFA formation and water movement. However, as stated in the answer to the first major review comment, our observational site lays on an ice divide with very little ice

dynamical changes reflected by a low mass change rate over the whole modelled period, so we judge these effects not significant enough

Furthermore, the source of non-coherence from towing speed and rough, possibly slightly different surface topography are to a first order represented by the difference in 2019 and 2018 data. Our observations show only slight differences, so we are confident in stating that the effect of differences in observational conditions (such as towing speed and slightly different topography) are small, and not enough the explain the error between modelled and observed water table depth in 2017, which is in the order of meters.

**Technical Corrections**
Fig 1: -  Significant place names should be added to the left hand side figure. Include label location of the Svalbard airport (as it is mentioned in the text). The figure caption is poorly worded and hard to understand.

We will add Longyearbyen Airport, and rewrite the figure caption.

I don't see 2015 or 2016 on the figure, only in the caption

We removed the years 2015 and 2016 because of their sparsity in useable data points, we will remove it from the caption

- 2017 looks green on the fig, indicated as yellow in the caption

Correct, a remnant of the deleted years from the previous comments. We will change this accordingly

- In sentence 2, its not clear what 'red rectangle' you are referring to – left or right hand figure.

Left hand side, we will change the color and the sentence to make it easier to distinguish.

For this sentence...."The rectangles correspond to the minimum and maximum coordinates of the measurement in those years and thus show the extend of the PFA measurements" .

- 'Extend' should be 'extent' (2 instances), turned should be converted,

We will change this

- Avoid over-use of first-person pronouns - it is very distracting to the readability of the paper. This is particularly an issue in the results section 4.1 and conclusion.

We will reduce the amount of first-person pronouns in the paper

- Please correct citations to use semi-colon, not comma to separate a list of references. An example of this is on line 50, but it happens throughout.

We will do this

- Please follow the rules of the journal when referring to figures and tables, noting the differences between use of these words at the beginning of the sentence versus in running text. Many errors of this kind throughout the paper.

We will read the guidelines of the journal and adjust the text accordingly

- Please note that the word "Table" is never abbreviated and **should be capitalized when followed by a number (e.g. Table 4).**

We will change this

- You have 'in situ' and 'in-situ' throughout, please be consistent with formatting rules.

We will use 'in situ'

L165: remove the word 'in' from the bracketed text.

We will do this

L215: 'transfer' doesn't seem like the right word… 'transform' or 'convert' perhaps?
   -remove space before comma

We will use 'convert' and remove the space before the comma

L238:  Improper use of brackets

We will remove 'e.g.' and write the list of studies as a 'normal' end-of-the-sentence reference. Also, we will change the comma to a semicolon

L216: change 'will' to 'is'
- It is not common for a DEM to have a variable cell size - please clarify.

This is a mistake on our side, the DEM we use has a constant cell size, which is 5 meters. We interpolate it to our grid, which has rectangular cells (72 x 96 meters) to accommodate 100 cells in every direction while maintaining a rectangular area around our observation locations. We will add this to the text.

L218: the differences here needs to be stated. please indicate rmsd between the datasets

We will rewrite this, this is from an older version of the model. We do not use the topography observations anymore: we simply use the depth to the water table directly from the GPR, and use the DEM from Melvaer et al (2014) to turn LPFAM output from height above sea level to depth to the water table.

L235: close off  should be close-off

We will change close off to close-off throughout the manuscript

L262: no hyphen in northwestern or southwestern

We will change this throughout the text

L263: include the max and min differences between the modelled and observed water table depths.

We will add this

L310; this sentence makes no sense on its own.

We are not sure to what sentence the reviewer is referring. The two sentences can be shortened and combined to: ' The water table depth and meltwater input at a single modelled grid cell, where the measurement station is located, is shown in Figure .... '

L325: should be Fig. not Fig

We will change this

L337: "*If refreezing happens, an ice lens might form*." This is a weird statement -  Why would ice not form if refreezing 'happens'?

We will change this to 'Meltwater refreezing creates an impermeable ice layer in the firn'

L349: improve the results by how much? Enough to explain the absence of crevasses as being the reason why coherence was so much lower?

We will provide the RMSE, and add a figure to the supplementary material in which we show the location of the added crevasses, and a discussion on the additional assumptions made.

L220-240: this section is misplaced as it describes methods, not results.

We agree and will move this part to the methods section

L252:  Change...

'*The modelled water table is overestimated in the south-eastern corner*'
 to
'*The modelled water table **depth** is overestimated in the south-eastern corner*'.

We will change this

L228: consequence is misspelt.

Thanks for noticing, we will change this

L320 and beyond, please refer to 'density' as 'firn density'

We will add this thourghout the manuscript.

Figure 2: y-axis – specify 'above mean sea-level'

We will change this

Figure 4. it would be far easier to compare the spatial pattern of differences between observed and modelled if the same color scheme for water table depth was used for both. Or, a third plot could be added to illustrate the differences.

We note that Figure 4 shows on the left side the modelled water table depth with the observation locations, and on the right already the difference between modelled water table depth and the observations. We see that the title of the right plots is misleading, it should be modelled water table depth difference wrt observations [year]. We will change this

It would be very useful here to have a box superimposed on the 2017 results which indicates the extent/positioning of the 2018/19 results.
Great suggestion, we will add this
- Caption refers to 'left column' and 'right'.
We will add 'right column'

Figure 5: units missing on both x and y axis.
We will add [m]
Figure 7:
- Fix caption (ie., Fig 7 to Figure 7.)
We will change this
- Color code left hand axis title to match blue line.
We will do that
- Refer to water table line as blue.
We will do this
- Not sure you need to specify both 'right' and 'orange'
We will only refer to 'orange' and 'blue'

**Comments by reviewer 2**

Transferred from PDF comments to word by Tim van den Akker. Answers are provided in blue.

Line 56-58: How is this model different from the model being used in this manuscript? Since this is the most recent model, please point out the detailed differences including assumptions and pros and cons of each approach.

Advantages include the combination of thermodynamics and firn modelling with the aquifer modelling in Miller et al (2023), and includes procedures to model unsaturated and saturated flow, a process that is missing in our approach. SUTRA-Ice is only tested on a 2D, flowline case of the Hellheim glacier with constant recharge rates. Our simulation is 3D, tested against three years of observations and including 'downscaled' climate input to simulate the behaviour of the aquifer over time.

We will add to Ln 57: The model SUTRA-ICE is comprehensive: it contains water flow through the unsaturated zone as well as water movement within the saturated zone. Freeze-thaw cycles are modelled, so a winter freeze of (a part of) the modelled PFA is represented. The model is tested on a 2D flowline of the Hellheim glacier, with constant recharge rates. 3D flow and realistic meltwater input from a (downscaled) climate or energy balance model is missing.

Fig 1: add ice edge boundary to this figure so the reader has a better sense of glacier setting. Also, make sure the colors of the rectangles match the caption and the years 2015 and 2016 do not appear in the text. Change 'next paragraph' to 'below'

We will remove the years 2015 and 2016, both from the figure as from the caption. We will change the colormap to make it easier to distinguish between height contours. We do not deem it nessecary to add glacier outlines to this graph, because in this study we do not refer to or use data from individual glaciers.

Line 111: Explain this in more detail. Are there any density vs depth measurements available?

Yes, snow pits have been dug at this location, to which the EBFM is tuned, see Van Pelt et al (2019). We then use a density profile of the EBFM close to the location of the measurements to calculate the dielectric constant for firn snow, which we use to calculate the velocity of the radar wave in the firn. We then use this velocity of the radar wave to turn the TWTT to the reflective surface that is semi-automatically picked to a depth to the water table. We will change Ln 111 – 114 to:

The raw GPR data was minimally processed with zero-time adjustment and a low-pass filter (300 MHz cut-off frequency). An example radargram is shown in Figure 2. The water table is picked from a radargram shown in Figure 2 semi-automatically: first, the reflective surface of the water table is manually found in a single data point. Then, a tracing algorithm is used to track that reflective surface through adjacent points. This results in  the two-way travel times (TWTT) per datapoint. Then, the velocity of the radar

wave in the firn is used to calculate the distance to the water table from the surface. For this, the dielectric constant of firn is required, which is calculated according to Eq 1 from Kovacs et al (1995) where $\rho f$ is the density of the firn layer and $\rho w$ the density of water:

The density of the firn at the location of the observations is obtained from the Energy Balance Firn Model (Van Pelt et al, 2019), of which a description is given in the next section.

Line 123: please give more details

We will rephrase this to: .. which has previously been calibrated and validated agains stake measurements, weather station data and observed density profiles from shallow firn cores. The location of these measurements are given in Van Pelt et al (2019) Section 2.3.

Line 140: is there a term that describes the density changes due to melt/freeze events?

Yes, the second F/dZ term in Eq 4. We will add to Line 140: , and F is the refreezing rate (in kg m$^{-2}$ s$^{-1}$). dZ is the layer thickness in m.

Line 219: can you quantify the difference?

Yes we can, but we rather not use the GPR measurements of the surface height. We will leave the GPR observations in terms of 'depth below the surface' , and transform LPFAM data, which is typically in height above sea level with the DEM from Melvaer et al (2014) to a depth below the surface. We will change Ln 211 – 2019 to:

Typical LPFAM output is the elevation of the water table in meters above sea level. The observational data is in water table depth below the surface. We use the Digital Elevation Model (DM) of Svalbard referred to as the Terrengmodel S0 with a resolution of 5 meters from Melvaer et al (2014), regridded to the model grid of the LPFAM as the firn surface. We then use the regridded DEM to subtract our modelled water table height above sea level from, to obtain the modelled water table depths from the LPFAM

Table 2: Can you estimate the uncertainty in this?

Yes, we can, and we will. We will add to Line 292:

There are uncertainties in the observed water table depth shown in this study. The system specific uncertainty (related to the sampling frequency, the cable length, and the GPR used) is small and about +- 0.02 meters. The largest source of uncertainty stems from the calculation of the velocity of the radar wave, which is calculated using a modelled density profile of the firn. When changing the firn density arbitrarily with +- 10%, this resulted in a spread of 0.21 m in calculated water table depths. The uncertainty arising from digitizing the water table, quantified by doing cross analysis of

double-measured points during the same field season, results in 0.03 m. Combined this gives an observational error of 0.26 m.

Fig 7: should this be blue?

Yes, thanks for spotting. We will change this.

Ln 354: Sentiel-1 may also be helpful and could potentially identify buried crevasses since C-band SAR can penetrate some snow covered surfaces

This is a great suggestion. We will add Sentinel 1 to our discussion and look for more prove on the existence of crevasses.

Ln 355: or potentially lakes depending upon local topography

We will add 'or meltwater lakes, if the surface topography allows for it'